# Securing Infrared Communication in Nuclear Power Plants: Advanced Encryption for Infrared Sensor Networks

**DOI:** 10.3390/s24072054

**Published:** 2024-03-23

**Authors:** Tae-Jin Park, Ki-il Kim, Sangook Moon

**Affiliations:** 1Nuclear System Integrity Sensing and Diagnosis Division, Korea Atomic Energy Research Institute (KAERI), 989-111 Daedeok-daero, Yuseong, Daejeon 34057, Republic of Korea; etjpark@kaeri.re.kr; 2Department of Computer Science and Engineering, Chungnam National University, Daejeon 34134, Republic of Korea; kikim@cnu.ac.kr; 3Department of Electrical and Electronic Engineering, Mokwon University, 88 Doanbuk-ro, Seo-gu, Daejeon 35349, Republic of Korea

**Keywords:** infrared communication, IoT sensors, data encryption algorithm, S-box security, nuclear power plants

## Abstract

This study enhances infrared communication security in nuclear power plants’ secondary systems, addressing the risk of mechanical and cyber failures. A novel random address generator, employing an innovative S-box, was developed to secure IoT sensor data transmissions to gateway nodes, mitigating eavesdropping, interference, and replay attacks. We introduced a structured IR communication protocol, generating unique, encrypted addresses to prevent unauthorized access. Key-dependent S-boxes, based on a compound chaotic map system, significantly improved encryption, increasing data transmission randomness and uniqueness. Entropy analysis and reduced duplicated addresses confirmed the effectiveness of our method, with the Hash-CCM algorithm showing the highest entropy and fewest duplicates. Integrating advanced cryptographic techniques into IR systems significantly enhances nuclear power plants’ security, contributing to the protection of critical infrastructure from cyber threats and ensuring operational integrity.

## 1. Introduction

Figure 1 illustrates the chronological progression of nuclear power plant (NPP) failures from 2017 to 2023 in South Korea [1]. It is observed that the total number of failures (represented by red dots) incurred from different sources shows a notable increase over time. With regard to managing NPPs, the advent of remote wireless diagnosis systems marks a pivotal shift towards enhancing human safety and operational efficiency. These systems are indispensable in addressing the challenges posed by aging infrastructure, where mechanical failures and radiation leakage risks are exacerbated by material degradation in high-temperature and radiation-exposed environments. This paradigm shift focuses attention on the necessity for secure and reliable communication networks, particularly tailored to the unique conditions prevailing within NPPs. Prominent applicable wireless technologies include Bluetooth, Zigbee, Wi-Fi, LoRa, WirelessHART, Terahertz, infrared (IR) communication, and more [2]. Among these technologies, IR, the electromagnetic spectrum of which falls between the red end of visible light and microwaves, offers distinct advantages in the context of monitoring nuclear steam supply systems [3]. While traditional wireless technologies offer a broad spectrum of solutions, the peculiar requirements of NPP environments—characterized by stringent regulatory standards and the critical need for minimal electromagnetic interference—demand a more focused approach. The hazard detection system using sensor networks in NPPs requires the minimum power possible to collect periodic data from thousands of sensors. In addition to this unusual type of sensor network environment, the system should comply with the regulations for radio-frequency interference (RFI) and electromagnetic interference (EMI) restrictions [4]. IR communication, with its inherent advantages of low power consumption and reduced electromagnetic interference, offers a compelling solution for such specialized applications. Taking that into consideration, the role of IR communication emerges as exceptionally significant.

However, the application of IR communication in NPPs is not without its challenges. The primary concern revolves around securing data transmission, considering the severe consequences of cyber-attacks on such vital infrastructure. Cyber-attacks, such as the Nordex wind turbine system shutdown in Germany in 2022, the Ukrainian power grid hack in 2015, and the Stuxnet worm in Iran, illustrate the potential threat of security breaches that can lead to unexpected termination of processes, malicious stealing of resources, and drastic damage to power-generating machines [5,6]. Recent incidents of cyber-attacks on power facilities globally accentuate the vulnerability of energy systems to security breaches, thereby amplifying the need for robust security measures. In light of this, our study presents an advanced encryption method specifically designed for IR sensor networks within NPP loops responsible for transferring heat and steam.

To address these challenges, this paper proposes a novel encryption strategy that employs a custom-designed S-box to secure data transmission from IR sensors to gateway nodes within NPPs. This strategy is developed to meet the unique security requirements of NPP operational loops, focusing on minimizing power consumption while maximizing data protection against cyber threats. By implementing this targeted encryption approach, we aim to enhance the security framework for IR communication in NPPs, ensuring the integrity and reliability of critical data transmission. This effort is expected to serve as a model for implementing similar security measures across various critical infrastructure sectors, thereby advancing the conversation on cybersecurity strategies in the face of growing digital threats.

## 2. Infrared Communication

The most commonly used protocol in IR communication is the NEC protocol, which is regarded as the de facto standard for IR data transactions [7]. NEC, a leading electronics company, has contributed significantly to establishing this protocol. In the NEC protocol, bit pattern encoding is governed by the spacing between pulses, with each bit’s representation being composed of a combination of pulse and space. The pulse operates at a 38 kHz frequency, leading to a fundamental period of 0.56 ms. A logical ‘1’—indicating a True condition—is represented by a structured sequence: pulse followed by long (three consecutive) spaces. This sequence occupies a total duration of 2.25 ms, as depicted in Figure 2. Conversely, a logical ‘0’, signifying a False condition, is denoted by a sequence of pulse followed by short (one single) space, resulting in an overall period of 1.125 ms, also illustrated in Figure 2.

When the message transmission starts, the following protocol is executed in order to establish the communication (see Figure 3). First, a 9 ms leading pulse burst is transmitted, followed by 4.5 ms space, then comes the next 8-bit slave device address for 27 ms duration, followed by its complemented values during 27 ms, then the 8-bit command (data) followed by the same length of duplicated values in complementary forms during 27 ms. The communication ends with the final 562.5 ms pulse. As shown in Figure 3, the data are transmitted in the order of least significant bits (LSBs) through most significant bits (MSBs).

## 3. Security Enhancement in Infrared Communication

### 3.1. Infrared Communication Security Issues

IR communication has security issues that can be exploited by malicious actors. One of the main security issues of IR communication is eavesdropping. IR signals are not directional, and they can be intercepted by anyone within the range of the signal. This means that an attacker can easily intercept and decode the data transmitted over the IR channel [8]. Another security issue of IR communication is interference. Since IR signals are light waves, they are susceptible to interference from other light sources. This can cause errors in the transmission of data, leading to data corruption or loss [9]. Moreover, IR communication can also be vulnerable to replay attacks. A replay attack involves an attacker intercepting and recording a legitimate IR signal and then replaying it to gain unauthorized access to a device or network [10]. To mitigate these security issues, various security measures have been proposed. Encryption can be used to secure the transmission of data over the IR channel. Additionally, directional IR signals can be used to limit the range of the signal and prevent eavesdropping [11]. 

### 3.2. Related Studies

Due to its inherent secure characteristics of spatial restrictions such as short distance limitation and the requirement of an obstacle-free environment, IR communication has seldom been used with encryption. In Akgul’s work in 2011 [12], authors proposed a new protocol and applied the Tiny Encryption Algorithm (TEA) symmetric cipher algorithm [13], but the reason for suggesting the new protocol and using the encryption algorithm was not sufficient. More recently, Kim and Suh focused on the structure of the IR transaction data packet and indicated that the packet can be divided into two sections: address and command (data) [14]. The authors applied secret key encryption for the IR data and suggested that rather than encrypting the entire message, partial encryption of only address section can be beneficial to save power and hardware area. They adopted the idea of a counter synchronization with a non-linear matrix to defend from the replay attack. However, the secret key pair exchange mechanism was not considered in depth, which is one of the most important factors to consider in implementing a secure communication system.

### 3.3. Rijndael S-Box Pair

The Rijndael S-box is a pair of substitution boxes, which is a critical component of the Advanced Encryption Standard (AES) cryptosystem [15]. Each instance of the forward S-box and inverse S-box serves an essential role during encryption and decryption, respectively.

***Forward S-box*** Forward S-box provides mapping for the function ***s*** = *S*(***c***) taking in 8-bit input to produce 8-bit output using the Galois Field (*GF*) residue system using *GF*(2^8^) = *GF*(2)[*x*]/(*x*^8^ + *x*^4^ + *x*^3^ + *x* + 1) [16]. The input c is transformed into its multiplicative inverse and applied to the matrix expression as shown below, where $\left[\begin{array}{cccccccc} s_{0} & s_{1} & s_{2} & s_{3} & s_{4} & s_{5} & s_{6} & s_{7} \end{array}\right]$ is the S-box output and $\left[\begin{array}{cccccccc} b_{0} & b_{1} & b_{2} & b_{3} & b_{4} & b_{5} & b_{6} & b_{7} \end{array}\right]$ is the multiplicative inverse of S-box input ***c***. According to the GF operation theory, + operation is implemented by the bitwise XOR operator, or ⊕.
(1)$$\left[\begin{array}{c} s_{0}\\ s_{1}\\ s_{2}\\ s_{3}\\ s_{4}\\ s_{5}\\ s_{6}\\ s_{7} \end{array}\right]=\left[\begin{array}{cccccccc} 1 & 0 & 0 & 0 & 1 & 1 & 1 & 1\\ 1 & 1 & 0 & 0 & 0 & 1 & 1 & 1\\ 1 & 1 & 1 & 0 & 0 & 0 & 1 & 1\\ 1 & 1 & 1 & 1 & 0 & 0 & 0 & 1\\ 1 & 1 & 1 & 1 & 1 & 0 & 0 & 0\\ 0 & 1 & 1 & 1 & 1 & 1 & 0 & 0\\ 0 & 0 & 1 & 1 & 1 & 1 & 1 & 0\\ 0 & 0 & 1 & 1 & 1 & 1 & 1 & 1 \end{array}\right]\;\left[\begin{array}{c} b_{0}\\ b_{1}\\ b_{2}\\ b_{3}\\ b_{4}\\ b_{5}\\ b_{6}\\ b_{7} \end{array}\right]+\left[\begin{array}{c} 1\\ 1\\ 0\\ 0\\ 0\\ 1\\ 1\\ 0 \end{array}\right]\;$$

***Inverse S-box*** Inverse S-box provides the reverse execution of the forward S-box. For example, if you look up the output of the Forward S-box when the input is 0 × 6*b*, which means 0 × 06 indicates the row and 0 × 0*b* the column, the output turns out to be 0 × 7*f*. Now, if the value 0 × 7*f* is applied as the input of the Inverse S-box, the row of 0 × 07 and the column of 0 × 0*f* is selected, which leads to the value of 0 × 6*b*. The calculation of contents of the inverse S-box conforms to the matrix expression as shown below. The input s is applied to the matrix expression as shown below, where $\left[\begin{array}{cccccccc} b_{0} & b_{1} & b_{2} & b_{3} & b_{4} & b_{5} & b_{6} & b_{7} \end{array}\right]$ is the multiplicative inverse of the Inverse S-box output and $\left[\begin{array}{cccccccc} s_{0} & s_{1} & s_{2} & s_{3} & s_{4} & s_{5} & s_{6} & s_{7} \end{array}\right]$ is the inverse S-box input ***s***.
(2)$$\left[\begin{array}{c} b_{0}\\ b_{1}\\ b_{2}\\ b_{3}\\ b_{4}\\ b_{5}\\ b_{6}\\ b_{7} \end{array}\right]=\left[\begin{array}{cccccccc} 0 & 0 & 1 & 0 & 0 & 1 & 0 & 1\\ 1 & 0 & 0 & 1 & 0 & 0 & 1 & 0\\ 0 & 1 & 0 & 0 & 1 & 0 & 0 & 1\\ 1 & 0 & 1 & 0 & 0 & 1 & 0 & 0\\ 0 & 1 & 0 & 1 & 0 & 0 & 1 & 0\\ 0 & 0 & 1 & 0 & 1 & 0 & 0 & 1\\ 1 & 0 & 0 & 1 & 0 & 1 & 0 & 0\\ 0 & 1 & 0 & 0 & 1 & 0 & 1 & 0 \end{array}\right]\;\left[\begin{array}{c} s_{0}\\ s_{1}\\ s_{2}\\ s_{3}\\ s_{4}\\ s_{5}\\ s_{6}\\ s_{7} \end{array}\right]+\left[\begin{array}{c} 1\\ 0\\ 1\\ 0\\ 0\\ 0\\ 0\\ 0 \end{array}\right]$$

The design of the S-box pair considers the concept of differential cryptanalysis, which is a type of attack that aims to determine the relationship between the input and output values of the S-box by analyzing the differences between pairs of input values [17]. To counteract this type of attack, the S-box is designed to have a high level of resistance to differential cryptanalysis, meaning that small changes in the input values result in large changes in the output values. Additionally, the design of the S-box considers the concept of algebraic immunity [18], which is a measure of the number of algebraic equations that are required to describe the mapping between the input and output values. A high level of algebraic immunity makes it difficult for an attacker to determine the relationship between the input and output values, thereby enhancing the security of the encryption. The purpose of the Rijndael S-box pair is based on a combination of mathematical concepts and principles, including the use of finite fields, non-linearity, resistance to differential cryptanalysis, and algebraic immunity. These criteria ensure the security and reliability of S-box permutation, making it useful for widely used encryption applications [19].

### 3.4. Proposed S-Box Secure IR Communication

A variety of strategies for the development of cryptographically effective S-boxes have been introduced. The contemporary techniques for S-box creation are primarily anchored in the chaotic mathematical theory. We introduce an innovative yet straightforward method for the construction of cryptographically strong key-dependent S-boxes, based on the compound chaotic map (which we henceforth call CCM) system [20]. The S-boxes devised here will perform an 8 × 8 mapping, thereby resulting in 16 × 16 matrices of S-boxes. Each S-box is comprised of a series of numbers between 0 and 255, building up a total of 256 possible configurations. Owing to the key-dependency of this S-box design methodology, a small change of even one bit in the key will lead to a completely different S-box. The pseudo code for generating new S-boxes is shown in detail (Algorithm 1):
**Algorithm****1: Enhanced S-Box Generation Using CCM**1. Initialize parameters:      - *x* and *r*, where *x* is in the range (0, 1) and 0 < *r* ≤ 9.      - iteration, within the bounds 105 ≤ iteration ≤ 109.      - Set *i* = 0, *m* = 0.      - Define an empty vector *S*.2. For *j* = 0 to iteration:      a. Check Loop Condition:           - If *i* > 255, then exit the loop (TerminateLoop).           - Otherwise, continue the loop.      b. Compute *x*:           - If *x* < 0.5, then *x* = (4 * sin(2π*xr*) + *x*) mod 1.           - Else, *x* = (4 * sin(2π*xr*) + (1 − *x*)) mod 1.      c. Calculate *m* = [(*x* * 10^14^) mod 256].      d. Validate *m*:           - If *m* is not in *S*, append *m* to *S* and increment both *i* and *j*.           - If m is in *S*, increment *j* only and repeat from step 2a.3. TerminateLoop:      - Return the *S* vector.

Table 1 summarizes the cryptanalytic analysis of CCM in reference to AES. In our study, we compare AES and CCM S-box methods across several security metrics, including non-linearity, the strict avalanche criterion (SAC), bit independence criterion—non-linearity (BIC-NL), bit independence criterion—strict avalanche criterion (BIC-SAC), linear probability (LP), and differential probability (DP). AES consistently demonstrates superior performance, signifying its robustness against linear and differential cryptanalysis with high scores in non-linearity, SAC, BIC-NL, and BIC-SAC, along with lower LP and DP values. These findings support AES’s effectiveness in ensuring unpredictability and resilience to cryptanalytic attacks.

While the CCM method exhibits commendable security attributes, it does not reach the high benchmarks set by AES in our analysis. The marginally higher LP and DP values for CCM suggest a comparative vulnerability to cryptanalysis. In response to this, our research introduces an innovative approach in Section 3.6 to augment the security properties of CCM for infrared communication applications. By implementing a feedback mechanism that reconfigures the seed value with each encryption cycle using the output from the previous S-box, we dynamically alter addresses to enhance security strength. This method addresses CCM’s inherent vulnerabilities by introducing added complexity and unpredictability into the encryption process, thereby significantly enhancing its defense against cryptographic threats.

### 3.5. Proposed Protocol for Secure IR Communication

Our target test bed for security enhanced IR communication is the secondary loop of a nuclear reactor, where thousands of IR sensors will be deployed to send the sensor data to receiver nodes that will subsequently pass it on to the cloud server. In the NEC protocol, the bit lengths for inverting address fields are only 8 bits, which are insufficient for connecting thousands of sensors, so we propose that we have 16-bit length address fields, theoretically enough for managing up to 2^16^ (65536) sensors. Other than the length of the address fields, we follow the de facto standard NEC protocol. This leads to an inevitable longer delay in the transmission time from 121.5 ms to 175.5 ms, as shown in Figure 4. Thanks to the peculiar structure of NEC’s duplicated inverting representation, the number of “1”s and “0”s should be the same, so the protocol enables easy synchronous implementations.

### 3.6. Self-Reconfiguring Pseudo Random Address Generator

We have adopted the idea of partially encrypting plain messages from [14], instead of encrypting the entire message, to save circuit area and reduce data transmission time. Additionally, we have adopted the non-linear S-box table pair (forward and inverse) proposed in Section 3.4. We consider the sensor nodes have 24-bit Network Interface Controller (NIC) field in their unique Media Access Control (MAC) address and extract 16-bits from NIC in two ways, which we explain later. Out of the 16-bit address fields, the higher 8-bit address goes through the forward S-box, and the lower 8-bit goes through the inverse S-box. We apply the generated S-box pair proposed in the above section and the Rijndael S-box pair as well for evaluation. Still, there exists a vulnerability unless the table mapping seed value has variable random characteristics. Figure 5 depicts the contents of proposed CCM and Rijndael S-box pairs in detail. The alphabet representation of the proposed S-box is capitalized to enhance the discriminability. 

In order to feed a new random seed every time one encrypted address is generated, we propose a simple but tricky structure, as shown in Figure 6. We note that S-boxes have an 8-bit interface, which allows us to construct 16 bits by combining results from two boxes. On iterations of every encryption, we XOR the upper 8 bits of the input address with the upper 8 bits of the encrypted address (initialized to 16’b0) and feed the result into the forward S-box. Similarly, we XOR the lower 8 bits of the input and encrypted addresses and feed the result into the inverse S-box. At this point, we add another encryption strength to the 16-bit S-box input vector, in two ways again. Firstly, as shown in Figure 6a, the top 16 bits of the NIC, which is the 24-bit part of the MAC that is the unique network address held by the edge sensor nodes, are hashed to create the upper 8 bits of input for the S-box, and the lower 16 bits are hashed to create the lower 8 bits of input for the S-box. The second method is simpler than using the hash function, as shown in Figure 6b, by supplying the initial input value for the S-box through XORing of the top and bottom 16 bits of the NIC. Now that we have two different S-box pairs and two different initialization methods, we have four variations to evaluate the performance. The 8-bit outputs of each S-box are concatenated to form a reliable random address. Crucially, each newly generated encrypted address is used as a seed value to generate the next address, thus ensuring continual reconfiguration of randomness and preventing vulnerability over time.

### 3.7. Implementation

We deploy this design of self-reconfigurability to build transceiver nodes for IR communication in our test bed. Figure 7 illustrates the security enhanced IR sensor network collecting data from the secondary loop in an NPP and transferring encrypted IR data to the receiver nodes. In each node of a pair of transceivers with CPUs and the gateway nodes as well, we equip the self-reconfiguring random address generator accordingly. IR communication is conducted between the transceivers according to the corresponding addresses, and every time data transaction is executed, the addresses are encrypted in a non-linear fashion with randomness so that external intruders cannot identify specific nodes or get control of any nodes. Since the time duration for one IR packet (address + data) is determined as 175.5 ms, synchronization is made naturally. This period is long enough for the CPUs embedded in the transceiver nodes to recalculate a new encrypted address.

As the software tool for the evaluation, we used the GNU Octave [21], which is open-source software that performs similar tasks to the commercial MATLAB, with comparable performance. We benefited from using Octave when dealing with thousands of address data by transforming them into encrypted data using empowered matrix operations. Octave also provides bit-manipulation operations, and we were able to utilize the dec2hex() and hex2dec() methods with matrices to effectively control the separation and combination of higher or lower bytes from or to 8-bit and 16-bit data. 

## 4. Evaluations 

Figure 8 shows the hypothetically expected outcome of our proposed random address generator. In this figure, we used the randi() function from the Octave tool library to obtain randomness while applying linearly incrementing 2000 test address values ranging from 0 to 1999. In Figure 8, the left-hand side figure depicts the straight line representing the linear address values. The right-hand side figure compares the linear values with the generated random values for clarity. We measured the strength of randomness by counting the number of duplicated address values. Since the randi() function generates different random numbers each time it is called, we repeated the process of generating 2000 random addresses 2000 times to ensure accuracy and consistency. The mean number of redundant address values was 30.492, and although the exact value is trivial, we can regard the number as greater than 30.

In the experiment, the original dataset comprises 2000 NIC address samples, each represented as a 24-bit integer. These integers represent a wide range of values, indicating a diverse dataset suitable for evaluating four proposed algorithm variations. Specifically, the dataset spans from a minimum value of 11,257 to a maximum of 16,758,366, with an average value of approximately 8,405,759.82. This broad distribution of values provides a robust basis for assessing the effectiveness and security of the proposed encryption techniques, ensuring a comprehensive evaluation across a variety of data characteristics. 

To assess the cryptographic strength of the proposed algorithms, an entropy analysis was conducted on the datasets generated by each algorithm. Entropy, a statistical measure of randomness that quantifies the uncertainty involved in predicting the value of a random variable, serves as a fundamental metric in evaluating the effectiveness and security of cryptographic systems. A higher entropy value indicates a higher level of unpredictability and randomness, thereby implying stronger encryption.

The entropy H of a dataset can be calculated using Formula (3): (3)$$H=-\sum_{i}p(x_{i})\log_{2}p(x_{i})$$
where *p*(*x*_*i*_) is the probability of occurrence of each unique value *x_i_* in the dataset. This method involves counting the occurrences of each unique encrypted value, calculating the probability of each unique value, and then applying the entropy formula to compute the overall entropy of the dataset [22].

### 4.1. XOR-AES Variation

Our first encrypted dataset *ra_dec_XOR_AES* exhibits a broad range of values, from a minimum of 5 to a maximum of 65,503, with an average value calculated at 32,783.60. This distribution suggests a uniform dispersion of encrypted values within the feasible range of 16-bit integers, which potentially indicates a level of randomness and unpredictability in the encrypted data. The comprehensive span of values, nearing the limits of the 16-bit integer capacity, hints at a deliberate utilization of the available numeric space, possibly contributing to the cryptographic strength of the algorithm. The mean value, closely aligning with the midpoint of the 16-bit integer range, further supports the hypothesis of an evenly distributed output, a desirable characteristic in encryption to mitigate patterns or biases that could be exploited for cryptanalysis. The entropy analysis of the *ra_dec_XOR_AES* encryption algorithm revealed an entropy value of 10.9368 bits. This indicates a high degree of randomness within the encrypted outputs, showcasing the algorithm’s effective obfuscation capabilities. The relatively high entropy value suggests that the *ra_dec_XOR_AES* algorithm provides robust security features, minimizing potential vulnerabilities to pattern-based attacks and enhancing the unpredictability of the encrypted data. 

For visualization, we compared the histogram of the original 24-bit NIC data against that of the encrypted output generated by the *ra_dec_XOR_AES* algorithm. Figure 9 reveals the effectiveness of the encryption process in obfuscating the original dataset’s distribution. The original data’s histogram, characterized by its distinct pattern and distribution, contrasts sharply with the more uniform and dispersed histogram of the encrypted data. This transformation highlights the *ra_dec_XOR_AES* algorithm’s capability to enhance data security by significantly increasing randomness and reducing predictability.

### 4.2. XOR-CCM Variation

*ra_dec_XOR_CCM* reveals from a minimum of 52 to a maximum of 65,511, with the mean value situated at 33,444.17. This range of values demonstrates the algorithm’s proficient employment of the 16-bit numeric field, ensuring comprehensive coverage that approaches the upper limits of possible values. The distribution pattern of the *ra_dec_XOR_CCM* dataset signifies a commendable degree of randomness, with the encrypted values spreading across the vast majority of the available space. This dispersion is indicative of the algorithm’s strategic encryption methodology, aimed at maximizing the unpredictability of the output. The average value, slightly elevated towards the upper end of the range, suggests a nuanced approach in the encryption process, potentially contributing to a higher level of security against cryptographic breaches. Furthermore, the close proximity of the maximum value to the theoretical limit of 16-bit integers underscores the algorithm’s ability to exploit the numeric space fully, enhancing the encrypted data’s resistance to pattern recognition and cryptanalytic attacks. The balanced and widespread distribution of values across the spectrum echoes the algorithm’s effectiveness in obfuscating the original data’s characteristics, a crucial aspect in maintaining data confidentiality. The entropy value associated with the *ra_dec_XOR_CCM* encryption algorithm was determined to be 10.9348 bits. This demonstrates the algorithm’s proficiency in ensuring a high level of randomness in its encrypted outputs, closely rivaling the performance of the *ra_dec_hash_CCM* algorithm. The *ra_dec_XOR_CCM* algorithm’s encryption outputs are characterized by a strong dispersal of values, indicating a robust defense mechanism against cryptographic breaches through pattern detection.

Further, the comparison between the original data and the *ra_dec_XOR_CCM* encrypted output through their histograms provides insight into the encryption’s impact on data randomness. The *ra_dec_XOR_CCM* algorithm alters the data distribution to achieve a more homogeneous spread of values, as opposed to the original data’s clustered distribution. This analysis accentuates the algorithm’s success in augmenting data unpredictability, a crucial attribute in safeguarding information from unauthorized access and analysis (Figure 10).

### 4.3. Hash-AES Variation

The third, *ra_dec_hash_AES*, showed a notable range of values, extending from a minimum of 60 to a maximum of 65,510, with an average value intricately positioned at 32,430.66. This distribution pattern underscores a well-distributed span of encrypted values, suggesting an effective dispersion across the entire 16-bit numeric spectrum. The examination of this dataset highlights the algorithm’s capability to exploit the available numeric space efficiently, thereby potentially enhancing the cryptographic strength through the promotion of randomness and unpredictability within the encrypted outputs. The proximity of the average value to the midpoint of the 16-bit range, coupled with the broad spread of values, supports the premise of an equitable and uniform distribution. For the *ra_dec_hash_AES* encryption algorithm, the computed entropy was 10.9288 bits. While this value reflects a substantial level of data randomness, it is marginally lower compared to the *ra_dec_XOR_AES* algorithm. This slight difference in entropy indicates a nuanced variation in the algorithm’s ability to distribute encrypted values across the available data space, which may influence the algorithm’s resistance to cryptanalytic attacks, albeit to a minor extent.

The analysis of our examination extends to the *ra_dec_hash_AES* encrypted data, compared alongside the original dataset’s histogram. The graphical representation illustrates a marked transformation in data distribution, from the original’s specific patterns to a flattened, more evenly spread distribution in the encrypted output. This comparison underscores the *ra_dec_hash_AES* algorithm’s proficiency in diluting original data characteristics, thereby fortifying the encryption strength against statistical analysis and pattern recognition (Figure 11).

### 4.4. Hash-CCM Variation

The final dataset, *ra_dec_hash_CCM*, exhibited a value range from a minimum of 3 to a maximum of 65,386, with the average encrypted value calculated to be 33,221.89. This analysis underscores the algorithm’s effective utilization of the 16-bit integer space, highlighting a substantial dispersion of values that encompasses nearly the entire potential range. The distribution characteristics observed in the *ra_dec_hash_CCM* dataset indicate a balanced and widespread allocation of encrypted values, suggesting a strategic approach to maximize randomness and reduce predictability within the encrypted data. Notably, the slight shift in the average value towards the higher end of the range, compared to previous algorithms analyzed, may reflect the algorithm’s unique encryption mechanism or data transformation strategy, which in turn could influence its resilience to cryptographic analysis. This dataset’s broad span of values and the near-central positioning of the average value illustrate the algorithm’s capacity to generate encrypted data that is evenly distributed across the available numeric spectrum. The *ra_dec_hash_CCM* encryption algorithm exhibited the highest entropy value among the evaluated algorithms, at 10.9438 bits. This superior entropy level underscores the algorithm’s exceptional performance in generating encrypted outputs with maximum randomness. Such a high degree of unpredictability significantly contributes to the algorithm’s cryptographic strength, providing enhanced security against attempts to decipher or predict the encrypted data.

Lastly, the *ra_dec_hash_CCM* algorithm’s encrypted data, when compared to the original, demonstrates a notable increase in distribution uniformity. The encrypted data’s histogram showcases a broad spread across the value range, diverging from the original data’s concentrated peaks. This evident dispersion serves as a testament to the *ra_dec_hash_CCM* algorithm’s effectiveness in obscuring data patterns, thereby elevating the security level and complicating potential decryption efforts without the key (Figure 12).

Lastly, the *ra_dec_hash_CCM* algorithm’s encrypted data, when compared to the original, demonstrate a notable increase in distribution uniformity. The encrypted data’s histogram showcases a broad spread across the value range, diverging from the original data’s concentrated peaks. This evident dispersion serves as a testament to the *ra_dec_hash_CCM* algorithm’s effectiveness in obscuring data patterns, thereby elevating the security level and complicating potential decryption efforts without the key.

## 5. Discussion

In our comprehensive evaluation of encryption algorithms, we analyzed the entropy values alongside the incidence of duplicated addresses produced by each algorithm. This analysis aimed to assess the randomness and uniqueness of the encrypted outputs, which are crucial for the overall security and integrity of the encrypted data. To calculate the entropy of each encryption algorithm’s output, we utilized the Shannon entropy formula, which is a measure of the unpredictability or randomness in a set of data. Specifically, the entropy was computed using the formula $H\left(X\right)=-\sum p\left(x_{i}\right)\log_{2}\left(p\left(x_{i}\right)\right)$, as shown in Formula (3). This method provided a quantitative assessment of the data complexity and unpredictability, with higher entropy values indicating greater randomness and, thus, stronger encryption.

Furthermore, we explored the relationship of our entropy calculation methodology with standards set forth by notable organizations such as the National Institute of Standards and Technology (NIST) and the Bundesamt für Sicherheit in der Informationstechnik (BSI) [23]. Both institutions emphasize the importance of entropy in assessing the strength of cryptographic systems. While our study did not directly apply the specific guidelines or methodologies recommended by NIST or BSI, our approach to entropy measurement aligns with the fundamental principles highlighted by these bodies. They advocate for the quantification of randomness as a critical factor in evaluating the security properties of encryption algorithms. By adhering to the universal principles of entropy calculation, our analysis ensures that the evaluation of encryption algorithms is grounded in well-established cryptographic standards.

The entropy values and the number of duplicated addresses indicative of each algorithm’s performance are shown in Table 2.

Our entropy analysis revealed that the Hash-CCM algorithm exhibited the highest entropy value at 10.9438 bits, indicating superior randomness in its encryption process compared to the other algorithms. High entropy is indicative of a robust encryption mechanism, as it reflects the unpredictability and complexity of the encrypted data. Furthermore, the number of duplicated addresses serves as an indicator of the uniqueness of the encrypted outputs. A lower incidence of duplicates suggests that the algorithm is more effective in generating distinct encrypted values, thus enhancing the security against potential cryptographic attacks. In this context, the Hash-CCM algorithm demonstrated the lowest number of duplicated addresses, with only 22 occurrences, underscoring its exceptional ability to maintain data uniqueness. Conversely, the Hash-AES algorithm, while still ensuring a high degree of security with an entropy value of 10.9288 bits, showed the highest number of duplicated addresses at 37. This suggests that, despite its effectiveness in randomizing data, there may be a slight reduction in the uniqueness of its outputs compared to Hash-CCM. The XOR-based algorithms, XOR-AES and XOR-CCM, displayed closely related entropy values and numbers of duplicated addresses, positioning them as reliable options for secure data encryption. However, they did not achieve the same level of performance in terms of both randomness and uniqueness as observed in the Hash-CCM algorithm.

Building upon the foundational analysis of entropy and duplicated address incidences, our further investigation through bit pattern change and correlation analyses has introduced new dimensions to our evaluation, particularly highlighting the efficacy of the Hash-CCM algorithm when combined with a self-reconfiguring approach and the CCM S-box. This combination not only adheres to the principles of robust encryption by enhancing data randomness and minimizing predictability but also surpasses the performance achieved with the conventional AES S-box usage.

The bit pattern change analysis, detailing the average frequency of bit alterations across the encryption process, and the correlation analysis, quantifying the degree of linear relationship between original and encrypted datasets, collectively affirm the superior encryption quality of Hash-CCM. These analyses demonstrate that Hash-CCM, especially when employing a self-reconfiguring system alongside the CCM S-box, introduces a higher level of data obfuscation compared to its counterparts. This assertion is backed by the statistical evidence presented earlier, where Hash-CCM not only yielded the highest entropy but also exhibited the lowest incidence of duplicated addresses, thus asserting its dominance in producing unique and unpredictable encrypted outputs. 

Figure 13 presents the distribution of 16-bit values generated by the encryption of 2000 samples of 24-bit NIC addresses using four distinct encryption algorithms: (a) XOR-AES, (b) XOR-CCM, (c) Hash-AES, and (d) Hash-CCM. Each subplot illustrates the variation in encrypted outputs, providing a visual representation of the algorithms’ effectiveness in producing a wide spread of values across the 16-bit space.

Upon thorough examination, which includes assessments of entropy values, incidences of duplicated addresses, bit pattern change frequencies, and correlation analyses, it is clear that the adoption of self-reconfiguring mechanisms alongside the CCM S-box in the Hash-CCM encryption algorithm surpasses the conventional cryptographic security benchmarks, notably outperforming the traditional AES S-box approach. This innovative approach makes Hash-CCM an exemplary model for encryption methodologies, particularly critical for enhancing IR communication security in specialized environments like remote diagnostics of NPPs.

The findings of our research emphasize the significance of evolving encryption strategies to strengthen communication security within critical infrastructure settings. The superior performance of Hash-CCM, characterized by its high entropy values, minimal duplicated addresses, significant bit pattern changes, and negligible correlation with the original data, renders it an optimal solution for securing IR communications. Such communications are vital for the remote monitoring and diagnostics of nuclear facilities, where data integrity and security cannot be compromised.

## 6. Conclusions

This research aimed to strengthen the security of infrared communication systems operating under the rigorous and sensitive conditions of nuclear power plants’ secondary loops. Our work was driven by the imperative to address the escalating vulnerabilities and preserve the operational continuity of aging nuclear infrastructures, as they weaken with heightened risks of mechanical and cyber failures. The heart of our exploration was the development and critical evaluation of a novel random address generator, designed to secure data transactions from IoT sensors to gateway nodes. Our proposed algorithms underwent rigorous scrutiny, assessed by entropy values to measure randomness and the number of duplicated addresses to determine uniqueness. The cryptographic assessment is meticulously summarized in Table 1, demonstrating the Hash-CCM algorithm’s superior encryption performance, marked by the highest entropy and the least duplicated addresses. Our findings, as visually explained in Figure 13, speak volumes about the proficiency of the Hash-CCM algorithm in enhancing security through data obfuscation and ensuring the non-repeatability of encrypted outputs. The algorithm’s pronounced uniform distribution of values across the 16-bit space and its minimal incidence of duplicates position it as the paramount choice for scenarios demanding stringent security measures. The XOR-AES and XOR-CCM variations, while exhibiting a substantial level of randomness, presented a discernible predictability that could potentially be exploited. The Hash-AES variation, despite its robust encryption, hinted at a slight reduction in output uniqueness, as evidenced by a greater number of duplicated addresses. In drawing our research to a close, the insights gathered from the entropy analysis and the observed distribution patterns underscore the imperative of selecting an encryption strategy that aligns with the unique security requirements of the application at hand. The Hash-CCM algorithm, with its outstanding performance, paves the way for secure, low-power, and short-range data transmission solutions, particularly in environments where manual monitoring poses a substantial challenge. Our work not only contributes a formidable encryption tool to the existing arsenal against cyber threats but also serves as a testament to the potential of IR communication in fostering a secure and resilient operational framework for nuclear power plants. As we address the intricate challenges of cybersecurity in essential infrastructure, the approaches and discoveries presented in this study will undoubtedly guide further progress and practical applications, ensuring that the security of our energy resources is maintained.

## Figures and Tables

**Figure 1 sensors-24-02054-f001:**
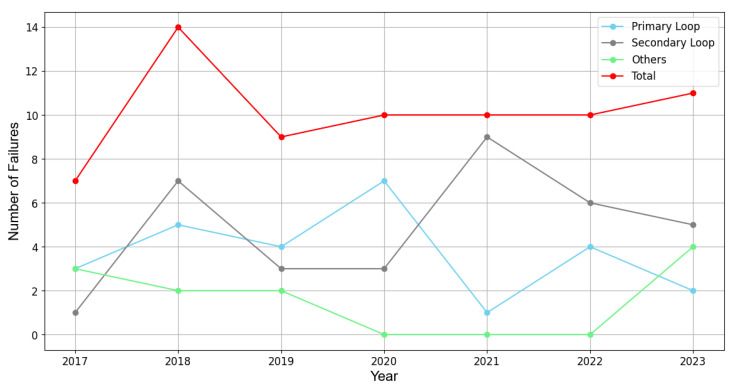
NPP fault sources and total number of faults from 2017 to 2023 in South Korea.

**Figure 2 sensors-24-02054-f002:**
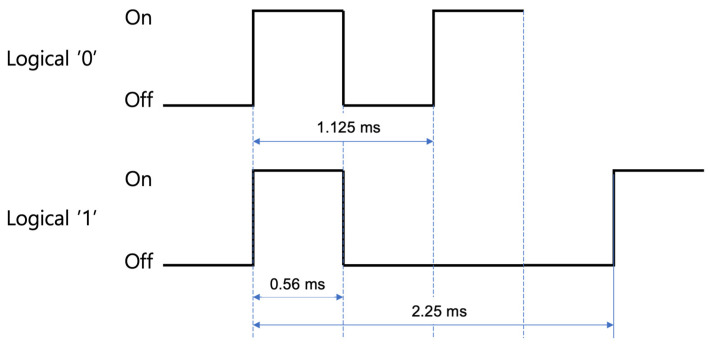
NEC data transmission protocol and the corresponding time duration.

**Figure 3 sensors-24-02054-f003:**
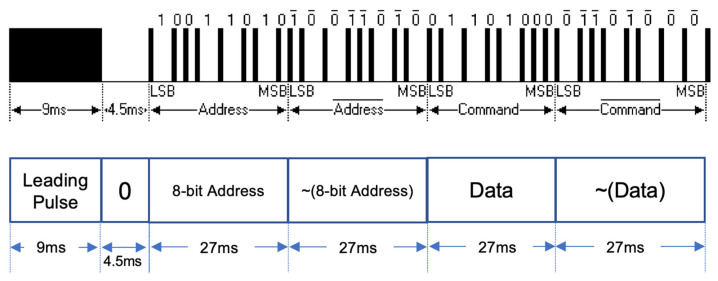
Pulse modulation structure for logical True (“1”) and False (“0”).

**Figure 4 sensors-24-02054-f004:**
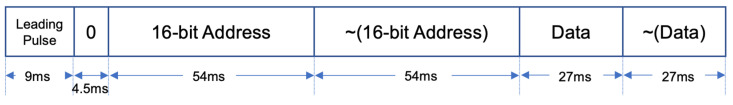
Proposed IR communication protocol with 16-bit address and 8-bit data.

**Figure 5 sensors-24-02054-f005:**
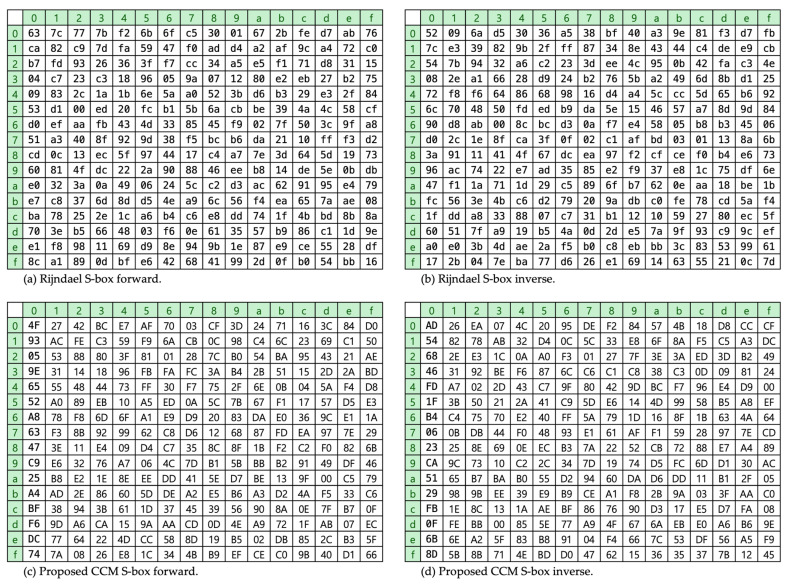
Contents of the S-box tables: (**a**) Rijndael S-box forward; (**b**) Rijndael S-box inverse; (**c**) CCM S-box forward; (**d**) CCM S-box inverse.

**Figure 6 sensors-24-02054-f006:**
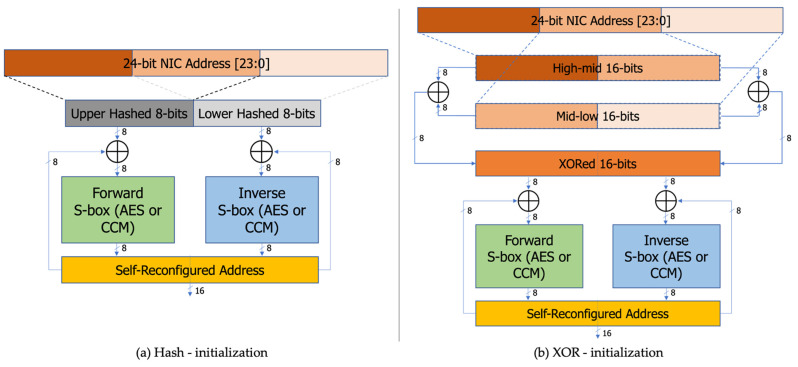
Proposed self-reconfiguring random address generator: (**a**) Hash-initialization with AES or CCM S-box; (**b**) XOR-initialization with AES or CCM S-box.

**Figure 7 sensors-24-02054-f007:**
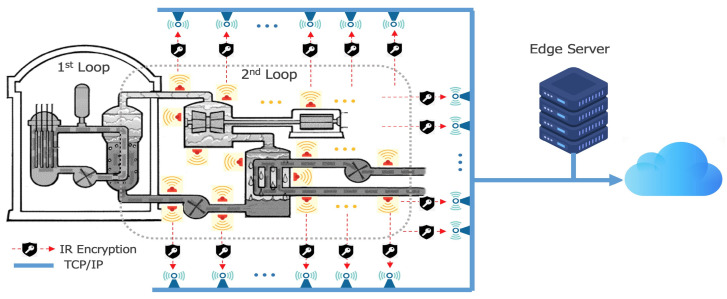
Sensors deployed in the secondary loop of a nuclear reactor as our test bed.

**Figure 8 sensors-24-02054-f008:**
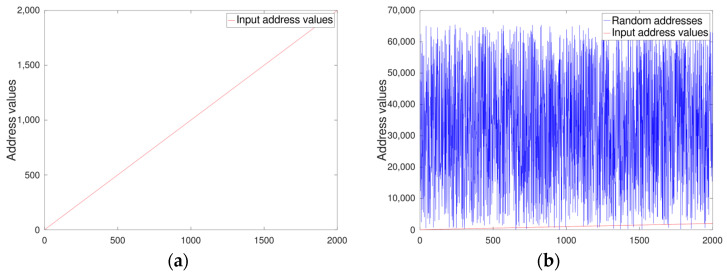
Randomness of pseudo random address values: (**a**) linearly increasing address values; (**b**) random address values generated by the randi() function in contrast with the red linear address values line at the bottom.

**Figure 9 sensors-24-02054-f009:**
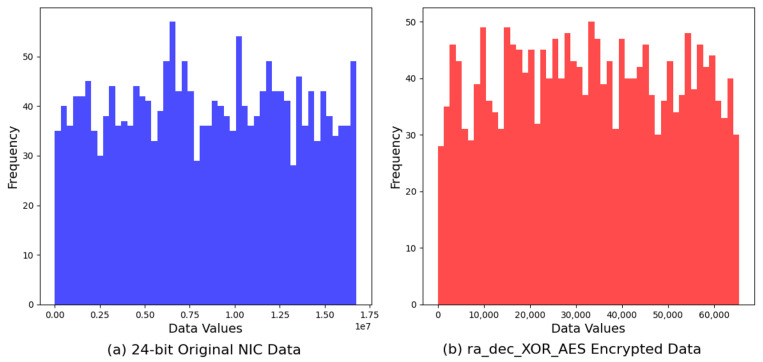
Comparison of original 24-bit NIC address vs. XOR-AES encryption.

**Figure 10 sensors-24-02054-f010:**
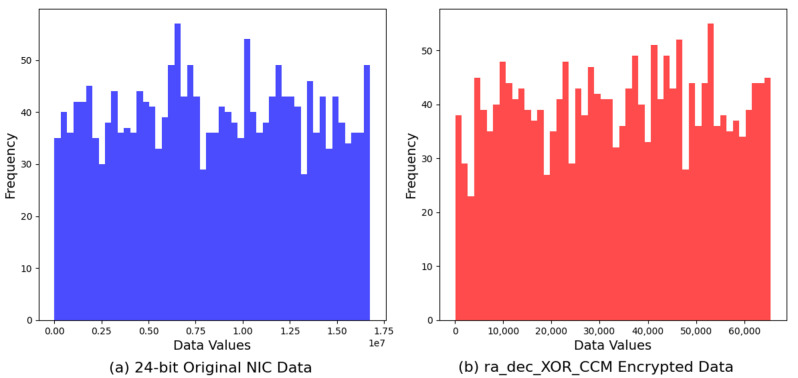
Comparison of original 24-bit NIC address vs. XOR-CCM encryption.

**Figure 11 sensors-24-02054-f011:**
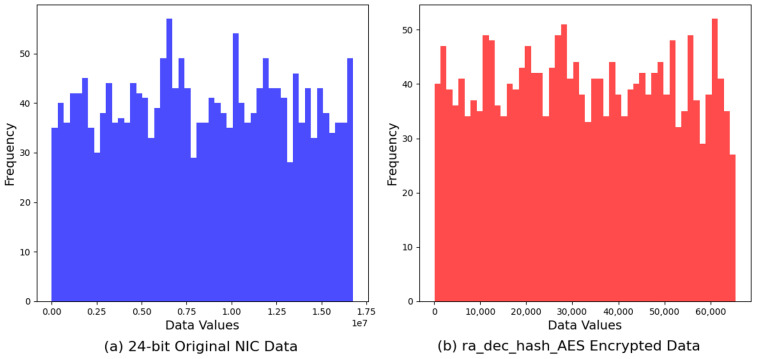
Comparison of original 24-bit NIC address vs. Hash-AES encryption.

**Figure 12 sensors-24-02054-f012:**
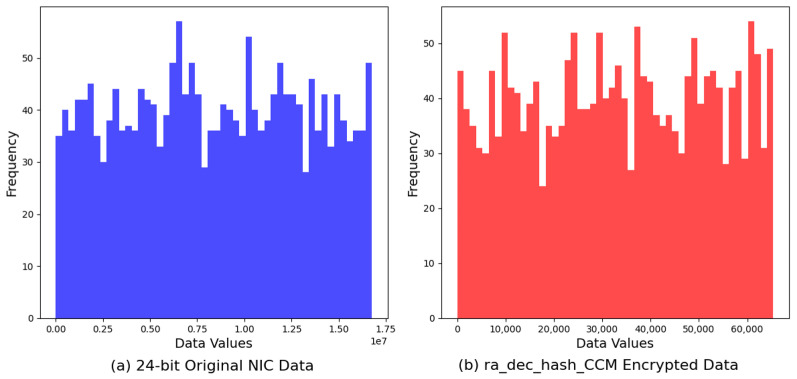
Comparison of original 24-bit NIC address vs. Hash-CCM encryption.

**Figure 13 sensors-24-02054-f013:**
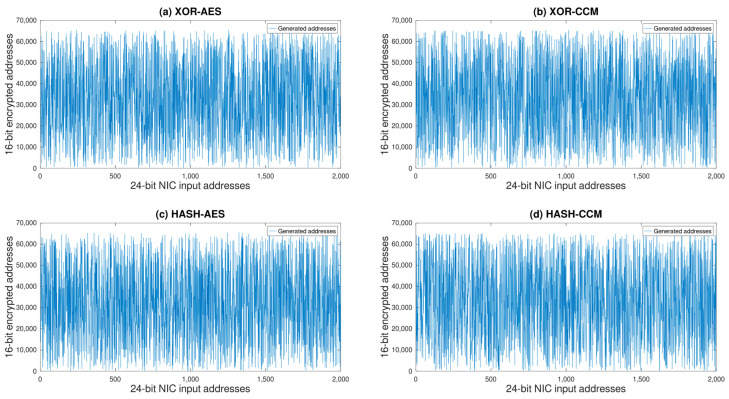
Comparison of four proposed encryption algorithms.

**Table 1 sensors-24-02054-t001:** Comparative analysis of AES and CCM S-box security metrics.

S-Box Method	Non-Linearity			SAC	BIC-NL	BIC-SAC	LP	DP
	Max	Min	Avg					
AES [20]	112	112	112	0.504	112	0.504	0.062	0.016
CCM	110	104	106.75	0.5027	103.6	0.5020	0.1328	0.0391

**Table 2 sensors-24-02054-t002:** Cryptographic assessment of four proposed encryption algorithms.

Algorithm	Average Bit Change Frequency	Correlation with Original	Entropy Value	# of Duplicated Addresses
XOR-AES	0.4986	0.0359	10.9368	29
XOR-CCM	0.4976	−0.0261	10.9348	30
Hash-AES	0.4985	−0.0292	10.9288	37
Hash-CCM	0.4978	0.0063	10.9438	22

## Data Availability

The datasets generated from the current study are available from the corresponding author upon reasonable request.

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
