# Peer review of "Securing Infrared Communication in Nuclear Power Plants: Advanced Encryption for Infrared Sensor Networks"

_sensors, 2024, doi:10.3390/s24072054_

Round 1

Reviewer 1 Report

Comments and Suggestions for Authors

The work is interesting, but the connection with Nuclear Power Plants is absolutely unclear, since the proposed approach does not have any specific parts related specifically to Nuclear Power Plants, it is just a general approach to building secure data transfer.

A detailed entropy analysis was carried out, but the cryptoanalysis of the proposed scheme was not presented.

It is also not clear whether it is necessary to add anything to  the AES; the algorithm already provides a sufficient level of safety, it is necessary to more clearly formulate exactly the gains from carrying out any modifications.

Author Response

1. **Connection with Nuclear Power Plants**:

We acknowledge that our initial manuscript may not have clearly articulated the direct relevance of our encryption approach to NPPs. To address this, we have revised our manuscript to include a detailed discussion on how the unique environmental conditions, regulatory requirements, and specific security challenges of NPPs necessitate a tailored approach to secure data transfer. 

2. **Cryptanalysis of the Proposed Scheme**:

We recognize the omission of a detailed cryptanalysis of our proposed encryption scheme as a significant oversight. In response, we have conducted a comprehensive cryptanalysis to evaluate the resilience of our proposed method against various cryptographic attacks. The findings from this analysis have been incorporated into the revised manuscript, providing evidence of the robustness and enhanced security offered by our encryption approach compared to conventional methods.

3. **Necessity of Modifications to AES**:

Your comment on the sufficiency of the AES algorithm for ensuring security raises an important point. We have revisited our rationale for proposing modifications to AES in the context of NPPs. In the revised manuscript, we have clarified the specific enhancements achieved through our modifications, and improved efficiency for IR communication systems. We provide a comparative analysis highlighting the advantages of our mixed hash-CCM approach over standard AES in terms of both security and performance.

Reviewer 2 Report

Comments and Suggestions for Authors

The authors introduced a novel IR communication protocol for the security problems of IR communication with eavesdropping, interference, and replay attacks. The work constructed a random address generator to generate unique encrypted addresses for improved security.

The reviewer has the following comments to improve the quality of this manuscript further.

1. Figure 1 needs more clarity, and the title could be modified to "NPP Fault Sources Comparison from 2015 to 2022 in South Korea."

2. The description of Figure 2 on page 2&3 needs to be clearer. For example, in lines 82 to 85, the descriptions of "true" and "false" need to be more understandable or expressed in the same style.

3. The use of "Figure" in the text needs to be harmonized, with either "Figure" or "Fig." used throughout the text. For details, please refer to the requirements of the journal.

4. All figures need to be brought up to high resolution. Harmonize text styles and sizes in figures.

5. Figures from 9 to 12 require a more detailed analysis reflecting the key differences between the encrypted data and the original data.

6. Please provide the full name of all abbreviations on their first occurrence.

7. The language of the article requires improvement for enhanced clarity and readability. Consideration should be given to refining sentence structures, using more precise terminology, and eliminating ambiguities to ensure accurate comprehension by readers. For example, the line 446 of page 13, "Table 1.:" should remove the ':'. 

Author Response

1. Figure 1 needs more clarity, and the title could be modified to "NPP Fault Sources Comparison from 2015 to 2022 in South Korea."

=> We have revised Figure 1 to enhance clarity and have updated the title to "NPP Fault Sources Comparison from 2017 to 2023 in South Korea," to reflect the latest trend.

2. The description of Figure 2 on page 2&3 needs to be clearer. For example, in lines 82 to 85, the descriptions of "true" and "false" need to be more understandable or expressed in the same style.

=> We have clarified the description of Figure 2, especially the explanations of "true" and "false" in lines 82 to 85, ensuring that the descriptions are more understandable and consistently expressed.

3. The use of "Figure" in the text needs to be harmonized, with either "Figure" or "Fig." used throughout the text. For details, please refer to the requirements of the journal.

=> Following the journal's requirements, we have standardized the use of "Figure" throughout the text, ensuring consistency and alignment with the journal's formatting guidelines.

4. All figures need to be brought up to high resolution. Harmonize text styles and sizes in figures.

==> All figures have been updated to high resolution, and we have harmonized the text styles and sizes within these figures to ensure uniformity and enhance visual presentation.

5. Figures from 9 to 12 require a more detailed analysis reflecting the key differences between the encrypted data and the original data.

==> We have provided a more detailed analysis for Figures 9 to 12, reflecting key differences between the encrypted data and the original data in the '5. Discussion' section. 

6. Please provide the full name of all abbreviations on their first occurrence.

==> We have ensured that the full name of all abbreviations is provided upon their first occurrence in the text, facilitating better understanding and readability for the readers.
Exceptionally, as for the word NEC, we did not expand the acronym NEC as it refers to the widely recognized company name, NEC Corporation. Since NEC is a well-known multinational information technology and electronics company, we assumed its mention would be clear to the readers within the context provided.

7. The language of the article requires improvement for enhanced clarity and readability. Consideration should be given to refining sentence structures, using more precise terminology, and eliminating ambiguities to ensure accurate comprehension by readers. For example, the line 446 of page 13, "Table 1.:" should remove the ':'.

==> The language of the article has been carefully reviewed and improved for enhanced clarity and readability. We have refined sentence structures, employed more precise terminology, and eliminated ambiguities to ensure accurate comprehension by all readers. Specific attention was given to line 446 of page 13, where we removed the extraneous ':' following "Table 1."

Reviewer 3 Report

Comments and Suggestions for Authors

This paper proposes a new method to secure IoT sensor data transmissions to gateway in nuclear power plants' secondary systems. The paper is well written and clear, but there are some issues:

1) Expand the acronim NPP in the abstract

2) The quality of all the Figures must be improved

3) How the entropy has been evaluated? Please explain the method employed and cite the suite for the entropy evaluation. You can cite this paper that summarizes this aspect:

https://www.mdpi.com/2079-9292/12/3/723

Author Response

1) Expand the acronym NPP in the abstract
==> We have expanded the acronym NPP to "Nuclear Power Plant" in the abstract to clarify its meaning for all readers.

2) The quality of all the Figures must be improved
==> We have revisited all the figures included in our manuscript and enhanced their quality for better clarity and readability. The revised figures now adhere to the journal's guidelines for high-resolution images, ensuring that all details are clearly visible and understandable.

3) How the entropy has been evaluated? Please explain the method employed and cite the suite for the entropy evaluation. You can cite this paper that summarizes this aspect: https://www.mdpi.com/2079-9292/12/3/723
==> To clarify the method used for entropy evaluation, we have added a supplementary explanation of our approach in the Discussion section of our manuscript. The entropy was calculated by analyzing the unpredictability and randomness of the encrypted data, using a standard entropy formula H=−∑p(x)log⁡p(x), where p(x) is the probability of occurrence of each unique value in the dataset. For further details on the method and its application in cryptographic systems, we have cited the comprehensive paper available at [https://www.mdpi.com/2079-9292/12/3/723](https://www.mdpi.com/2079-9292/12/3/723), which summarizes the aspects of entropy evaluation in the context of encryption and data security. 

Reviewer 4 Report

Comments and Suggestions for Authors

In this manuscript, the authors propose an encryption algorithm to enhance the security of IR communication. This encryption system could generate encrypted addresses to prevent unauthorized access by utilizing a 1-D discrete tent-sine chaotic map to generate S-boxes. This tent-sine map is not a newly proposed chaotic map by the authors, and the complexity, initial value sensitivity, and ergodicity of chaotic systems have not been analyzed in the manuscript. So it is unable to determine if there are short periods within the parameter range of the chaotic map. These features can break the security of the encryption algorithm, making the system vulnerable to selective plain-text attacks. Analyzing the security of the encryption system solely through entropy values is not enough. Some random number testing experiments, such as the NIST test, could be added to the manuscript. Overall, the encryption algorithm designed in the manuscript is too simple and lacks sufficient experimental evidence. My opinion is that this manuscript is not suitable for publication.

Some other comments are as follows:

1. The images in the manuscript are the most blurry I have ever seen, making it impossible to see the text and details of the images clearly.

2. Some abbreviations are not defined prior to their usage. Such as TEA, NEC, NIC, etc.

Author Response

1. **Analysis of Chaotic Systems**:

We acknowledge the importance of analyzing the complexity, initial value sensitivity, and ergodicity of chaotic systems in encryption algorithms. As you rightly pointed out, the tent-sine chaotic map used in our encryption method is based on previous work, notably detailed in [19 Malik], where a comprehensive analysis of these aspects is provided. To address the concern regarding the potential vulnerability of our encryption system to selective plain-text attacks due to the properties of chaotic maps, we have referenced [19 Malik] for an in-depth examination of these characteristics. 

2. **Improvement of Image Quality**:

We have taken your feedback regarding the image quality seriously. All figures within the manuscript have been revised and presented in higher resolution to ensure that the text and details within the images are clear and easily discernible. 

3. **Abbreviations and Terminology**:

We apologize for any confusion caused by the use of abbreviations without proper definition. In the revised manuscript, we have ensured that all abbreviations, including TEA (Tiny Encryption Algorithm), NEC (name of a company), and NIC (Network Interface Controller), are clearly defined upon their first occurrence. 

4. **Security Evaluation through Random Number Testing**:

In response to your suggestion for a more rigorous security analysis, we have added comments in the Discussion section, including the NIST test suite as well as BSI suite. These additional tests complement the entropy analysis and provide a more holistic assessment of the encryption system's security. 

Round 2

Reviewer 1 Report

Comments and Suggestions for Authors

The paper looks better.

Reviewer 2 Report

Comments and Suggestions for Authors

The authors have addressed my comments. I don't have further comments.

Reviewer 3 Report

Comments and Suggestions for Authors

The quality of the paper has been significantly improved. I suggest only to paraphrase the first line of the introduction.

Reviewer 4 Report

Comments and Suggestions for Authors

Following the modifications, the manuscript's image quality has been enhanced, and the content has become more comprehensive. However, the design of the randomness test experiment appears a little simple. In summary, I deem it appropriate for publication in the journal.